# mTORC1 is required for differentiation of germline stem cells in the *Drosophila melanogaster* testis

**Marie Clémot**[1,2], **Cecilia D'Alterio**[1], **Alexa C. Kwang**[1], **D. Leanne Jones**[1,2,3,4,5]*

**1** Department of Molecular, Cell and Developmental Biology, University of California, Los Angeles, Los Angeles, CA, United States of America, **2** Eli and Edythe Broad Center of Regenerative Medicine and Stem Cell Research, University of California, Los Angeles, Los Angeles, CA, United States of America, **3** Departments of Anatomy, Division of Geriatrics, University of California, San Francisco, San Francisco, CA, United States of America, **4** Departments of Medicine, Division of Geriatrics, University of California, San Francisco, San Francisco, CA, United States of America, **5** Eli and Edythe Broad Center for Regeneration Medicine, University of California, San Francisco, San Francisco, CA, United States of America

* leanne.jones@ucsf.edu

**Data Availability Statement:** All relevant data are within the manuscript.

**Funding:** Funds to support this study were provided by the UCLA Eli and Edythe Broad Center of Regenerative Medicine and Stem Cell Research

## Abstract

Metabolism participates in the control of stem cell function and subsequent maintenance of tissue homeostasis. How this is achieved in the context of adult stem cell niches in coordination with other local and intrinsic signaling cues is not completely understood. The Target of Rapamycin (TOR) pathway is a master regulator of metabolism and plays essential roles in stem cell maintenance and differentiation. In the *Drosophila* male germline, mTORC1 is active in germline stem cells (GSCs) and early germ cells. Targeted RNAi-mediated downregulation of mTor in early germ cells causes a block and/or a delay in differentiation, resulting in an accumulation of germ cells with GSC-like features. These early germ cells also contain unusually large and dysfunctional autolysosomes. In addition, downregulation of mTor in adult male GSCs and early germ cells causes non-autonomous activation of mTORC1 in neighboring cyst cells, which correlates with a disruption in the coordination of germline and somatic differentiation. Our study identifies a previously uncharacterized role of the TOR pathway in regulating male germline differentiation.

## Introduction

Adult stem cells have the unique ability to both self-renew and produce daughter cells that acquire a distinct identity through differentiation, thereby playing essential roles in the maintenance of tissue homeostasis. A myriad of factors has been shown to contribute to the regulation of stem cell behavior, which include cell-intrinsic factors and extrinsic cues, either systemic or provided by the local microenvironment ("niche") [1–3]. In particular, the capacity of stem cells to self-renew or differentiate can be attributed to distinct metabolic states; consequently, metabolism has emerged as an important regulator of stem cell fate decisions [4–8]. Yet, how metabolic cues are integrated with other signaling pathways to influence stem cell behavior in the niche is not fully understood.

Training Program (MC), and the NIH: AG040288, AG052732, GM135767 (DLJ): The funders had no role in study design, data collection and analysis, decision to publish, or preparation of the manuscript.

**Competing interests:** The authors have no competing interests

The testis of *Drosophila melanogaster* is an excellent system for investigating the mechanisms that regulate stem cell behavior. Furthermore, as it contains two populations of adult stem cells—germline stem cells (GSCs) and somatic cyst stem cells (CySCs), the testis provides an ideal model to address questions related to how multiple stem cell systems can be coordinated to maintain tissue homeostasis [9, 10]. Recently, a genetic screen conducted in our laboratory, designed to uncover the role of mitochondrial dynamics in regulating *Drosophila* male GSC behavior, showed that the mitochondrial fusion protein Mitofusin (dMfn) is essential for GSC maintenance [11]. Depletion of dMfn in early germ cells resulted in ectopic phosphorylation of 4E-BP, a substrate of mTor kinase, in these cells, suggesting that the Target of Rapamycin (TOR) pathway becomes hyperactivated in the absence of dMfn. Consistent with this observation, the TOR inhibitor rapamycin partially rescued GSC loss induced by dMfn depletion in early germ cells [11]. Therefore, TOR hyperactivity appears to contribute to GSC loss in the *Drosophila* testis upon perturbation of mitochondrial fusion in germ cells, raising the question of the role of TOR in the regulation of adult male GSCs under homeostatic conditions.

The TOR pathway is a highly conserved signaling network that functions as a metabolic rheostat, coordinating cellular growth and proliferation with the availability of resources [12–14]. The Ser/Thr kinase mTor is the catalytic core of the pathway and forms part of two distinct complexes: mechanistic target of rapamycin complex 1 (mTORC1) and mechanistic target of rapamycin complex 2 (mTORC2), which differ in accessory proteins, upstream regulatory signals and substrate specificity. The mTORC1 complex controls metabolic pathways in response to intracellular nutrients and energy levels, as well as extracellular factors such as hormones and growth factors, thereby integrating local and systemic cues. Upon activation, mTORC1 promotes anabolic processes, including protein, lipid and nucleotide synthesis, while repressing catabolic processes such as autophagy, whereas mTORC2 regulates cell survival and proliferation [15]. Given its central role in metabolism, the TOR pathway provides a tool to gain insights into how metabolic inputs are integrated with other signaling pathways to regulate stem cell fate decisions.

Indeed, mTORC1 regulates stem cell behavior and fate decisions across many systems, including murine spermatogonial progenitor cells, mesenchymal stem cells, hematopoietic stem cells, intestinal stem cells, and hair follicle stem cells, among others [16]. Interestingly, mTORC1 appears to be required for both stem cell maintenance and differentiation of progenitor cells, suggesting that mTORC1 activity must be fine-tuned and tightly regulated [16].

The genetic toolkit available in *Drosophila* allows for modulation of the TOR pathway in a temporal and tissue-specific manner, facilitating analysis of the role of TOR in adult stem cell maintenance, rather than during development. For example, in the *Drosophila* ovary, TOR was found to be required for GSC maintenance and proliferation, as well as for the growth and survival of differentiating progeny [17]. By contrast, hyperactivation of mTORC1 through cell-specific disruption of its repressor Tuberous sclerosis complex (TSC1/2), causes precocious GSC differentiation [18]. In addition, precisely tuned mTORC1 activity controls the number of mitotic divisions and the mitotic to meiotic transition in germline cysts [19, 20]. In the *Drosophila* testis, a role for mTORC1 in somatic cyst cell differentiation has been described [21, 22]; however, the role of TOR in adult male GSCs under homeostatic conditions has not been addressed.

Here, we describe mTORC1 activity in male GSCs and early germ cells. Depletion of *mTor* in early germ cells leads to an accumulation of germ cells with GSC-like features, reflecting a block or delay in differentiation. Germ cells depleted for *mTor* kinase activity also harbor unusually large and dysfunctional autolysosomes. Furthermore, downregulation of *mTor* in adult GSCs and early germ cells leads to a striking non-autonomous activation of mTORC1 in

neighboring cyst cells, which correlates with precocious differentiation. Our study highlights a previously uncharacterized role of the TOR pathway in male germ cells, expanding what is known about this crucial regulator of metabolic and tissue homeostasis.

## Materials and methods

### Fly stocks and genetics

The *Drosophila* transgenes used in this study are listed in **Table 1** and all fly stocks were raised on standard cornmeal medium.

The *UAS-RFP* sensor was generated by subcloning the *RFP-PASS4E* coding sequence, encoding an inactive phosphatidic acid (PA) sensor unable to bind PA [24] (plasmid kindly provided by Guangwei Du, University of Texas) into a pUAS-attP plasmid, immediately downstream of the UAS sequence, by InFusion cloning. Transgenic lines were generated by Bestgene (Chino Hills, CA) through injection of the resulting plasmid into a strain containing an attP40 site.

Early adult specific depletion in early germ cells using the *nanos^{TS}* driver was achieved by raising flies at 18˚C and shifting the adult progeny to 29˚C, for either 7 days (*Tor^{RNAi1}*) or 15 days (*Tor^{RNAi2}*) to induce expression of the RNAi transgenes. We noted that this system had variable efficiency depending on the RNAi transgenes used. For example, expression of *raptor^{RNAi}* in adult germ cells using *nanos^{TS}* did not cause any reduction in pS6 staining (not shown), indicating that Raptor is likely not depleted at a sufficiently high level to cause a reduction in TOR activity, whereas its expression throughout development using *nanosGAL4:VP16* efficiently led to a loss of pS6 in germ cells (**S4D Fig**).

Five RNAi transgenes targeting different regions of *mTor* mRNA were tested (listed in **Table 1**). *mTor^{RNAi1}*, *mTor^{RNAi2}* and *mTor^{RNAi3}* caused the strongest phenotypes when expressed in germ cells throughout development with *nanosGAL4:VP16*. *mTor^{RNAi4}* caused a milder phenotype and *mTor^{RNAi5}* did not cause any obvious phenotype. Thus, *mTor^{RNAi1}*, *mTor^{RNAi2}* and *mTor^{RNAi3}* were expressed in adult germ cells only using *nanos^{TS}*. *mTor^{RNAi1}* and *mTor^{RNAi2}* induced a robust phenotype and an efficient reduction in mTORC1 activity, assessed by pS6 staining, after 7 days and 15 days of induction at 29˚C, respectively. By contrast, *mTor^{RNAi3}* did not result in efficient reduction of pS6, even after 15 days of induction.

Transgene expression in germ cells throughout development was achieved using *nanos-GAL4:VP16* or *bam-GAL4:VP16* at 25˚C.

**Rapamycin feeding.** Rapamycin feeding was performed as previously described [21]. 100µl of a 4mM Rapamycin (TSZ Chem, Cat# R1017) stock solution in ethanol was added to the surface of standard diet food and air dried. Similarly, control food was prepared by adding 100µl of ethanol to the surface of standard diet food. Flies were transferred to fresh rapamycin- or ethanol- treated food every 2–3 days, for the period of time indicated for each experiment.

**Immunostaining.** Adult testes were dissected in PBS and fixed in 4% paraformaldehyde for 30 minutes, then permeabilized by two 15 minutes washes in 0.3% Sodium Deoxycholate and washed for 15 minutes in 0.1% PBT (1x PBS, 0.1% Triton X-100). Tissues were incubated in 3% Bovine Serum Albumine (BSA)-0.1% PBT blocking solution for at least 30 minutes. Primary antibodies (listed in **Table 1**) were diluted in blocking solution as follow: chicken anti-GFP (1:1000), mouse anti-α-Spectrin (1:20), mouse anti-Fas3 (1:100), rabbit anti-Vasa (1:100), rat anti-vasa (1:25), guinea pig anti-TJ (1:3000), mouse anti-Eya (1:20), mouse anti-LaminC (1:20), rabbit anti-phospho-dRpS6 Ser 233/235/239 (1:200), anti-phospho-dRpS6 Ser 233/235 (1:400), rabbit anti-phospho-4E-BP1 Thr37/46 (1:500), rabbit anti-phospho-SMAD1/5 Ser463/465 (1:100), rabbit anti-phospho-Stat92E (1:50), mouse anti-phospho-histone H3 (1:300), mouse anti-Bam (1:10), rabbit anti-Ref(2)P (1:200), rat anti-DE-Cad (1:20) and mouse

**Table 1. List of fly stocks and reagents.**

| REAGENT | SOURCE | REFERENCE |
|---|---|---|
| *Drosophila* transgenes | | |
| mTor::GFP | Vienna Drosophila Resource Center (VDRC), Fly TransgeneOme (fTRG) collection | Stock #318201 |
| UAS-RFP | this publication | |
| bam-GAL4:VP16 | Laboratory of Margaret Fuller, Stanford University | |
| nanos-GAL4 | Bloomington Drosophila Stock Center (BDSC) | Stock #4442 |
| tub-GAL80$^{ts}$ | BDSC | Stock #7018 |
| tub-GAL80$^{ts}$ | BDSC | Stock #7108 |
| $y^1,w^1$ | BDSC | Stock #1495 |
| UAS-GFP.nls | BDSC | Stock #4775 |
| nanos-GAL4:VP16 | BDSC | Stock #4937 |
| UAS-mTor$^{TRiP-HMS00904}$ (referred to as RNAi1) | BDSC, Transgenic RNAi Project (TRiP) collection | Stock #33951 |
| UAS-mTor$^{TRiP-HMS01114}$ (referred to as RNAi2) | BDSC, TRiP collection | Stock #34639 |
| UAS-mTor$^{shRNA}$ (referred to as RNAi3) | VDRC, shRNA collection | Stock #330679 |
| UAS-mTor$^{TriP-GL00156}$ (referred to as RNAi4) | BDSC, TRiP collection | Stock #35578 |
| UAS-mTor$^{dsRNA}$ (referred to as RNAi5) | Fly Stocks of National Institute of Genetics (NIG-FLY) | NIG 5092R-2 |
| UAS-iml1$^{TRiP-HMC04806}$ | BDSC, TRiP collection | Stock #57492 |
| UAS-white$^{TRiP-HMS00045}$ | BDSC, TRiP collection | Stock #33644 |
| UAS-raptor$^{TRiP-HMS00124}$ | BDSC, TRiP collection | Stock #34814 |
| UAS-thor | BDSC | Stock #9147 |
| pAtg8a-mCherry-Atg8a | Laboratory of Thomas Neufeld, University of Minnesota | |
| pAtg8a-GFP-mCherry-Atg8a | Laboratory of David Walker, University of California, Los Angeles | Lee at al., 2016 |
| tub-GFP:LAMP1 | Laboratory of Helmut Krämer, UT Southwestern | Pulipparacharuvil 2005, Akbar 2009 |
| **Antibodies** | | |
| Chicken polyclonal anti-GFP | Aves Labs | Cat #GFP-1020 |
| Mouse monoclonal anti-α-Spectrin (3A9) | Developmental Studies Hybridoma Bank (DSHB) | |
| Mouse monoclonal anti-Fasciclin 3 (Fas3) (7G10) | DSHB | |
| Rabbit polyclonal anti-Vasa (d-260) | Santa Cruz Biotechnology | Cat #sc-30210 |
| Rat monoclonal anti-Vasa | DSHB | |
| Guinea pig polyclonal anti-Traffic Jam (TJ) | Laboratory of Dorothea Godt, University of Toronto | Li et al., 2003 |
| Mouse monoclonal anti-Eyes absent (Eya) (10H6) | DSHB | |
| Mouse monoclonal anti-Lamin C (LC28.26) | DSHB | |
| Rabbit polyclonal anti-phospho-dRpS6 (Ser233/235/239) (pS6) | Laboratory of Aurelio Teleman, German Cancer Research Center | Romero-Pozuelo et al., 2017 |
| Rabbit polyclonal anti-phospho-dRpS6 (Ser233/235) (pS6) | Laboratory of Jongkyeong Chung, Seoul National University | Kim et al., 2017 |
| Rabbit monoclonal anti-phospho-4E-BP1 (Thr37/46) (236B4) | Cell Signaling Technology | Cat #2855 |
| Rabbit monoclonal anti-phospho-SMAD1/5 (Ser463/465) (41D10) (pMad) | Cell Signaling Technology | Cat #9516 |
| Rabbit polyclonal anti-Ref(2)P | Laboratory of David Walker, UCLA | Rana et al., 2017 [23] |
| Rat monoclonal anti-DE-Cad (DCAD2) | DSHB | |
| Mouse monoclonal anti-Patched (Ptc) (Apa 1) | DSHB | |
| Mouse monoclonal anti-phospho Histone H3 (Ser10) (6G3) | Cell Signaling Technology | Cat #9706 |
| Rabbit polyclonal anti phospho-*Drosophila* Stat92E (Tyr711) | Cell Signaling Technology | Cat #9357 |
| Mouse monoclonal anti-Bam | DSHB | |
| **Chemicals** | | |
| Rapamycin | TSZCHEM | Cat #R1017; CAS#53123-88-9 |
| Lysotracker Red | Invitrogen | Cat # L7528 |
| Click-iT$^{TM}$ EdU Cell Proliferation kit for imaging, Alexa Fluor$^{TM}$ 555 dye | Invitrogen | Cat #C10338 |

anti-Patched (1:500). Of note, the two anti-phospho-dRpS6 antibodies were initially tested in the testis and, as they produced similar patterns, they were later used interchangeably in the experiments presented in the manuscript. Following overnight incubation at 4°C with primary antibodies, samples were washed with PBT three times for 15 minutes, incubated in Alexa-conjugated secondary antibodies (Invitrogen, 1:500 in blocking solution), washed with PBT three times for 15 minutes and mounted in Vectashield® (Vector laboratories) with DAPI.

For Lysotracker Red staining, testes were dissected in PBS and incubated for 30 minutes in Lysotracker Red solution (in 1:1000 in PBS) prior to fixation in 4% paraformaldehyde for 20 minutes and continuation with the immunostaining protocol as described above, maintaining the tissues in the dark.

For EdU incorporation and labeling, testes were dissected in Schneider's *Drosophila* medium and incubated for 30 minutes in Schneider's *Drosophila* medium supplemented with 10μm EdU. Testes were then fixed in 4% PFA in PBS for 15 minutes, washed with PBS, permeabilized in 0.5% PBT for 20 minutes and Click-IT Edu labeling was performed according to the kit manufacturer's instructions, followed by blocking and immunostaining as described above.

**Microscopy and image analysis.** Fluorescent images were acquired with a Carl Zeiss Axio Vert.A1 inverted light microscope using a 63x oil-immersion objective. Images were processed using ZEN digital imaging and analyzed with the ImageJ software. All images shown are representative images of at least three independent experiments, with 20 to 24 testes per condition.

Quantitative experiments were evaluated for statistical significance using the software Graph-pad Prism v6.0. Graphics display means with standard deviations and the statistical differences between control and test samples were assessed using Mann-Whitney tests for discontinued variables such as number of cells or unpaired t-tests for continued variables. Statistical significance is denoted as * for $p < 0.05$, ** for $p < 0.01$, *** for $p < 0.001$, **** for $p < 0.0001$ and n.s (not significant) for $p > 0.05$. The number of values in each condition is indicated in the figure legends.

## Results

### The TOR pathway is active in male GSCs and early germ cells

As a first step to investigate the role of TOR signaling in the adult male germline, we assessed its activity. The adult *Drosophila* testis is a coiled tube, in which spermatogenesis proceeds in a well-defined spatiotemporal manner. The stem cell niche is located at the apical tip and contains GSCs (**Fig 1A**, circled in yellow), surrounding a group of somatic support cells referred to as the hub (**Fig 1A**, asterisk). The hub is a critical signaling center that coordinates the maintenance of both GSCs and CySCs. GSCs divide asymmetrically to self-renew and give rise to a gonialblast (GB). GBs are displaced away from the hub and undergo four rounds of synchronous transit amplification (TA) divisions with incomplete cytokinesis, thereby generating a group of 16 interconnected spermatogonia. These further differentiate into spermatocytes that undergo meiosis to form spermatids and mature sperm [25, 26]. Each GSC is surrounded by a pair of CySCs (**Fig 1A**, arrowheads), which self-renew and give rise to cyst cells (CCs). The somatic CCs enclose GBs and provide signals necessary for germ cell maturation [27]. Specific gene expression patterns and markers can be used to identify both stem cell pools and differentiating daughters along the length of the testis (**Fig 1A**) [9].

In order to determine whether TOR is active in adult germ cells, we took advantage of antibodies that specifically recognize phosphorylated *Drosophila* ribosomal protein S6 (dRpS6). Immunostaining with these antibodies was demonstrated previously to provide an accurate readout of TOR activity *in situ* in larval imaginal discs [28, 29]. In adult testes, phosphorylated dRpS6 (hereafter referred to as pS6) was detected in a subset of GSCs and early germ cells (**Fig 1B**). We validated pS6 as a reporter of TOR activity in the germline by feeding flies rapamycin,

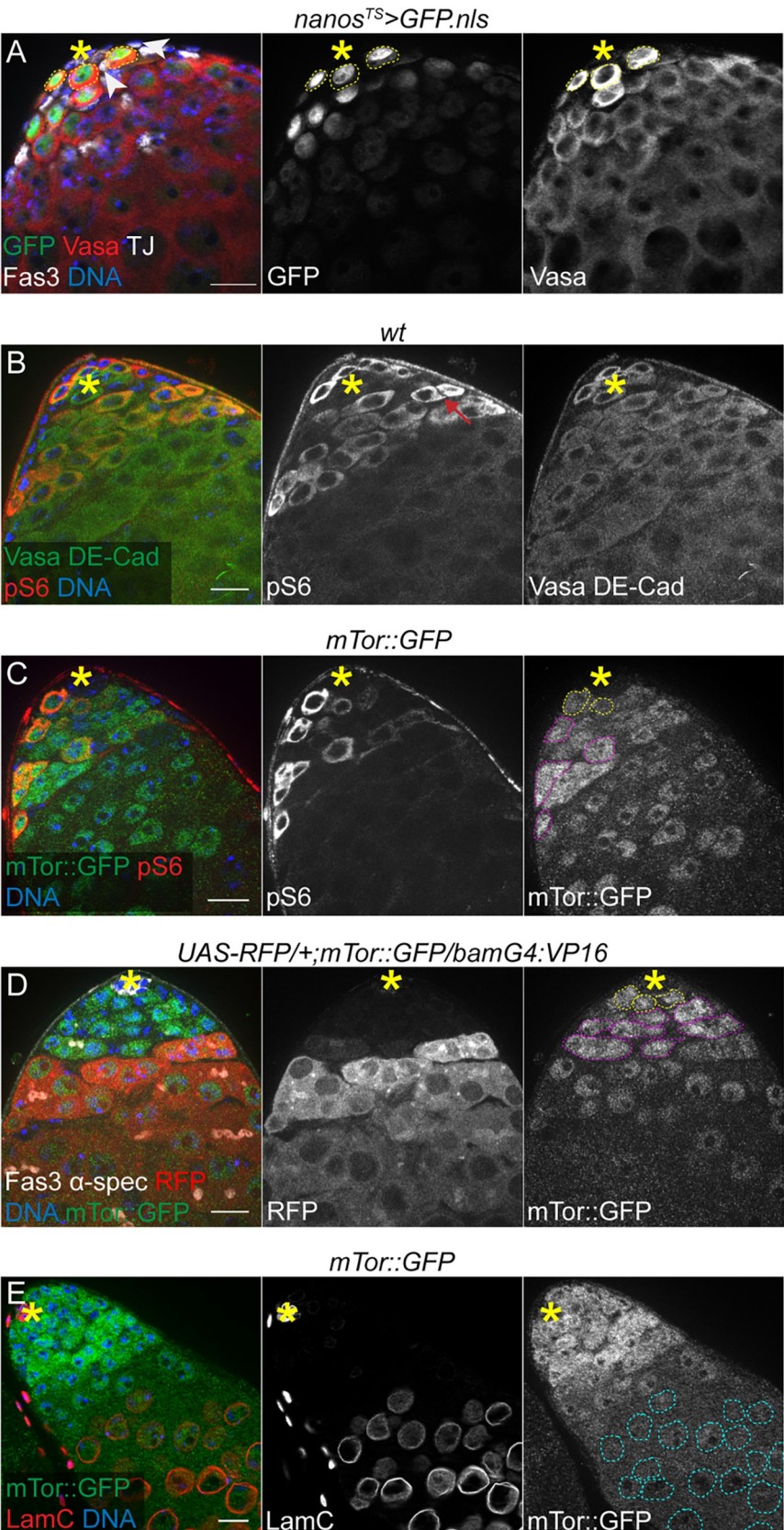

**Fig 1. The TOR pathway is active in GSCs and early germ cells.** (A) Cell identity markers in the testis stem cell niche. Germ cells are labeled with Vasa staining, hub cells (asterisk) with Fas3 and Traffic Jam (TJ) staining and early cyst cells with TJ staining, in a testis tip expressing nuclear GFP under the control of the early germ cell driver *nanos^{TS}*, upon 5 days of induction at 29˚C. GSCs (yellow dotted circles) are identified as Vasa$^+$ germ cells in contact with the hub and CySCs (arrowheads) are identified as TJ$^+$ cells in contact with the hub. (B) Phospho-dRpS6 (pS6) staining in the tip of wild-type (*y^1*,*w^1*) testis. The red arrow points to an example of spermatogonial cyst positive for pS6. E-Cadherin (DE-Cad) is used as a marker of hub cells. (C) pS6 staining in the tip of a *mTor::GFP* testis. pS6$^+$ GSCs (yellow) and spermatogonia (magenta) are circled in dotted lines in the right panel showing mTor::GFP expression alone. mTor::GFP is visualized by immunostaining with an anti-GFP antibody, as in the subsequent panels. (D) mTor:: GFP expression pattern in a testis tip expressing RFP under the control of *bam-GAL4:VP16*. α-spectrin (α-spec) marks the fusome in germ cells. GSCs, in contact with hub cells, are circled in yellow and early spermatogonia that do not express RFP are circled in magenta. (E) mTor::GFP expression pattern in a testis tip co-stained with LaminC (LamC). LamC is expressed at high levels in hub cells (asterisk) and in spermatocytes (cyan dotted circles). In all panels, asterisks indicate the location of the hub at the apical tip of the testis. Scale bars: 15μm.

a potent mTORC1 inhibitor [30]. Upon rapamycin feeding, pS6 staining strongly decreased in testes, indicating that pS6 is dependent on mTORC1 activity (**S1A, S1B and S1G Fig**). In addition, pS6 levels appear to reflect genetically induced changes in mTORC1 activity in germ cells. Depletion of mTor in GSCs and early germ cells induced a strong decrease in pS6 staining (**S1C, S1D and S1G Fig**), whereas depletion of Iml1, a component of the GATOR1 complex that represses mTORC1 activity [31, 32], induced a marked increase in the number of pS6-positive germ cells (**S1E and S1F Fig**). Interestingly, another validated target of TOR, phosphorylated 4E-BP (p4E-BP), was not detected in germ cells by immunostaining (**S1H Fig**), even upon genetically induced upregulation of TOR activity (**S1I Fig**). Overexpression of 4E-BP in germ cells, however, led to a pattern of p4E-BP similar to pS6, suggesting that 4E-BP is not normally expressed at high levels in early germ cells under homeostatic conditions (**S1J Fig**), which is further supported by scRNA-Seq data from the Fly Cell Atlas [33].

In addition, we observed that an *mTor::GFP* transgene, consisting of a large genomic clone containing *mTor* coding and regulatory sequences with a GFP tag inserted at the C-terminus [34], was differentially expressed in germ cells and coincided with TOR activity (**Fig 1C**). Cells at the tip of the testis, including GSCs (**Fig 1D**, yellow circles) and early spermatogonia (**Fig 1D**, magenta circles), exhibit higher levels of mTor::GFP, which then progressively decrease in cells expressing *bag of marbles* (*bam*) (**Fig 1D**), an early marker of germ cell differentiation. Spermatocytes, identified as germ cells expressing high levels of Lamin C [35], have very low to undetectable mTor::GFP expression (**Fig 1E**, cyan circles). Importantly, RNAi-mediated knock down of *mTor* in germ cells efficiently reduced GFP signal in these cells, confirming that GFP fluorescence largely reflects the expression of GFP-tagged mTor (**S1K and S1L Fig**). This is in agreement with the single-cell RNA sequencing data from the Fly Cell Atlas project that indicate that average *mTor* RNA levels decrease as germ cells enter the spermatocyte stage [33]. A similar pattern was observed in early female germ cells as they progress through differentiation during oogenesis (**S1M Fig**) [36]. Published data indicating that this *mTor::GFP* transgene does not rescue the larval lethality associated with a loss of function *mTor* allele [34] suggest that this transgene does not represent the complete pattern of endogenous *mTor* expression, which is evident by the lack of GFP expression in the somatic cells of the testis (**Fig 1C and 1D**). Altogether, our data indicate that TOR is active in GSCs and early germ cells in the testis, in addition to having a well-established role in the somatic cyst lineage [21, 22].

## mTor is required in adult GSCs and early germ cells for germline differentiation

In the adult testis, mTORC1 appears active in subsets of early germ cells (**Fig 1B and 1C**). In order to assess the role of mTORC1 activity in supporting germline homeostasis, we used

RNA interference (RNAi) combined with the bipartite UAS/GAL4 system [37] to specifically deplete mTor kinase in GCSs and early germ cells. The *nanos-GAL4* 'driver' promotes expression in GSCs and early germ cells (**Fig 1A**) [38]; when combined with a ubiquitously expressed thermo-sensitive *GAL80^TS* allele [39] (hereafter referred to as *nanos^TS*), this system permits depletion specifically in adult GSCs and early germ cells upon a shift to 29˚C (cf. Methods).

Induction of *mTor^RNAi* under the control of *nanos^TS* led to a significant decrease in mTor:: GFP expression (**S1L Fig**) and a decrease in pS6 staining specifically in Vasa⁺ germ cells (**S1C, S1D and S1G Fig**), confirming efficient depletion of mTor and subsequent loss of TOR activity in these cells. Conditional depletion of mTor kinase in adult male GSCs and early germ cells resulted in sterility and thinner testes (**Fig 2A and 2B**). Germ cells were still present at the tip of the testis (**Fig 2C and 2D**) and we observed a modest, but significant, decrease in the number of Vasa⁺ germ cells in direct contact with the hub (from 9.5±0.25, n = 39 in control testes to 7.1± 0.31, n = 46, upon mTor depletion, (**S2C Fig**). In order to determine whether mTor-depleted germ cells were able to differentiate, we assessed the distribution pattern of α-spectrin to highlight the fusome, a membranous structure that is spherical in GSCs and GBs and forms a branched structure as spermatogonia undergo TA divisions [40, 41]. Depletion of mTor in adult GSCs/early germ cells caused an accumulation of germ cells with a spherical fusome (**Fig 2C–2E**), suggesting that decreased mTor may lead to an arrest or delay in germ cell differentiation, prior to TA divisions. Alternatively, differentiating germ cells could de-differentiate, as has been described previously [42]. Importantly, expression of other RNAi transgenes targeting different regions of *mTor* mRNA and depletion of the mTORC1-specific subunit Raptor in GSCs and early germ cells throughout development also resulted in an accumulation of germ cells with a spherical fusome (**Fig 2E**, **S2A, S2B and S2G–S2H Fig**).

To further characterize the impact of mTor depletion on germ cell differentiation, Bone Morphogenetic Protein (BMP) pathway activity was assayed by phosphorylation of Mothers against Dpp (Mad) [43, 44]. The BMP pathway is normally activated in GSCs by Dpp and Gbb ligands, secreted by somatic hub cells and CySCs, which regulate germline differentiation [45–47]. Depletion of mTor led to a dramatic increase in the number of germ cells positive for phosphorylated Mad (pMad) (**Fig 2F–2H**). The intensity of pMad staining in mTor-depleted germ cells was markedly higher than in control germ cells (**Fig 2F and 2G**), and pMad⁺ germ cells were observed several cell diameters away from the hub. An accumulation of pMad⁺ germ cells was also observed upon 1) mTor depletion with another, independent RNAi line (**Fig 2H**; **S2I and S2J Fig**), 2) feeding flies with rapamycin (**S2K and S2L Fig**), and 3) depletion of the mTORC1-specific subunit Raptor in GSCs and early germ cells throughout development (**Fig 2H**; **S2M and S2N Fig**). Taken together, these data support our observations that depletion of mTor results in accumulation of early germ cells, likely GSCs/GBs, and suggest that mTORC1 negatively regulates BMP signaling in early germ cells. In agreement, we also observed that depletion of mTor in adult early germ cells caused a strong decrease in expression of the early differentiation marker Bam, which is negatively regulated by BMP signaling (**S2O and S2P Fig**).

Activation of the JAK/STAT pathway is another hallmark of GSCs [48, 49]. Accordingly, phospho-STAT, a readout for JAK/STAT activation, is detected in the nucleus of Vasa⁺ germ cells in contact with the hub (**S2S Fig**). Upon depletion of mTor in adult early germ cells, we observed an increase in the number of pSTAT⁺ germ cells (**S2T and S2U Fig**), further indicating that the single germ cells that accumulate in this condition have GSC-like features. Furthermore, whereas control spermatogonial cells divide in synchrony with the other cells of the cyst (**S2Q Fig**), single dividing germ cells can be seen several cell- diameters away from the hub upon mTor-depletion (**S2R Fig**). Therefore, mTor-depleted early germ cells retain the ability to proliferate in a GSC-like manner.

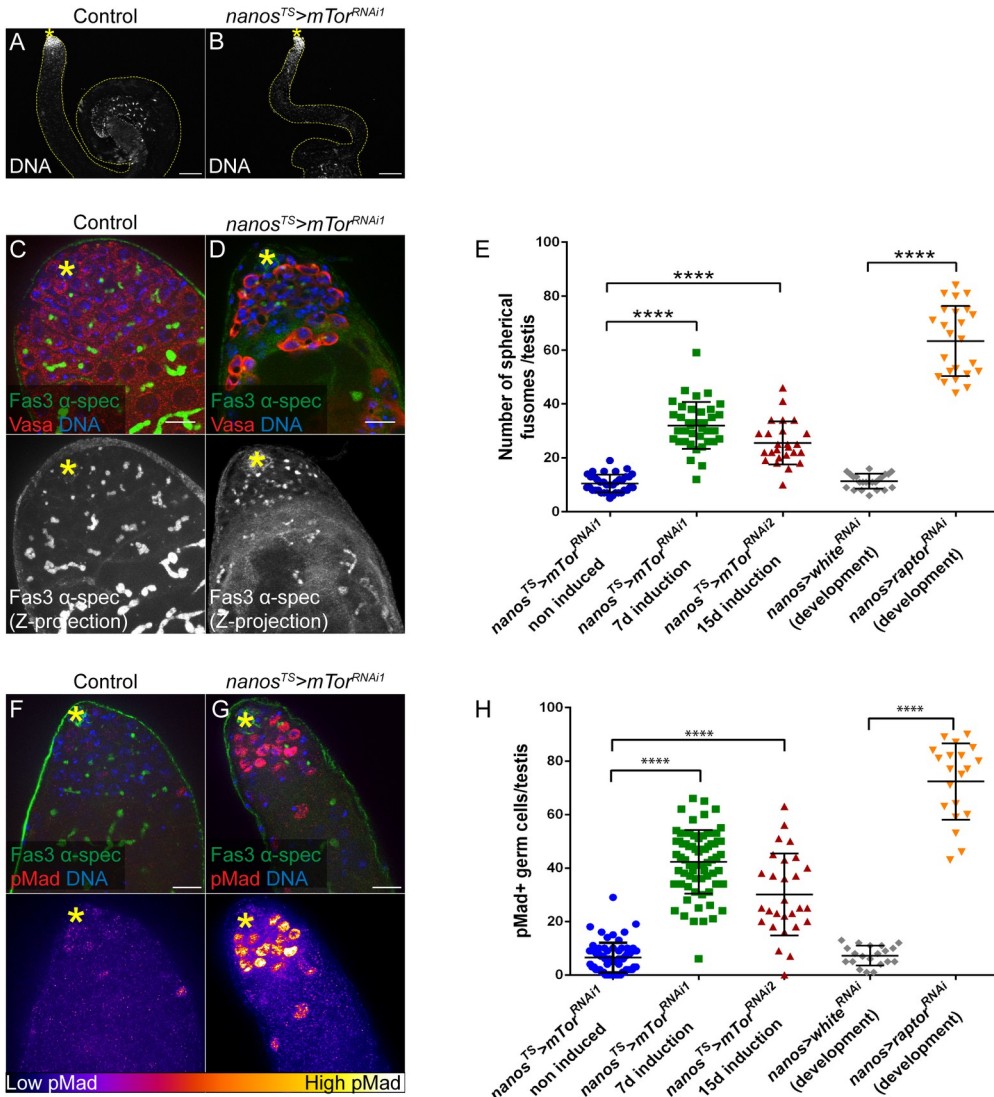

**Fig 2. Downregulation of mTORC1 in early germ cells causes an accumulation of GSC-like germ cells.** (A-B) Overall structure of single testes of the genotype *nanos-GAL4/+;UAS-mTor<sup>TRiP-HMS00904</sup>/tub-GAL80<sup>ts</sup>*, visualized by staining nuclei with DAPI (DNA) and contoured with yellow lines in non-induced control (flies maintained at 18˚C, A) and upon early germ cell-specific depletion of mTor (flies reared at 18˚C and shifted to 29˚C at the adult stage for 7 days to induce mTor depletion in GSCs and early germ cells, B). Scale bar: 100μm. (C-D) Testis tips of non-induced (control, C) or mTor-depleted (D) *nanos-GAL4/+;UAS-mTor<sup>TRiP-HMS00904</sup>/tub-GAL80<sup>ts</sup>* flies with Vasa, α-spectrin (α-spec) and Fas3 stainings. Lower panels show maximum intensity projections of several images spanning the hub region (Z-projections). Scale bar: 15μm. (E) Number of germ cells with a spherical fusome per testis of the indicated genotypes: *nanos<sup>TS</sup>>mTor<sup>RNAi1</sup>*—non induced (n = 37), *nanos<sup>TS</sup>>mTor<sup>RNAi1</sup>*–7 days induction (n = 37), *nanos<sup>TS</sup>>mTor<sup>RNAi2</sup>*–15 days induction (n = 25), *nanos>white<sup>RNAi</sup>*—expressed throughout development (n = 23) and *nanos>raptor<sup>RNAi</sup>*—expressed throughout development (n = 24). Means and standard deviations are shown. \*\*\*\* denotes statistical significance (p<0.0001) as determined with Mann-Whitney tests. (F-G) Testis tips of non-induced (control, F) or mTor-depleted (G) *nanos-GAL4/+;UAS-mTor<sup>TRiP-HMS00904</sup>/tub-GAL80<sup>ts</sup>* flies with phospho-Mad (pMad), α-spectrin and Fas3 stainings. Lower panels show pMad-associated fluorescence intensity represented as a color gradient with the lowest intensity in black and the brightest intensity in white. Scale bar: 15μm. (H) Number of pMad-positive germ cells per testis tip of the indicated conditions: *nanos<sup>TS</sup>>mTor<sup>RNAi1</sup>*—non induced (n = 66), *nanos<sup>TS</sup>>mTor<sup>RNAi1</sup>*–7 days induction (n = 71), *nanos<sup>TS</sup>>mTor<sup>RNAi2</sup>*–15 days induction (n = 28), *nanos>white<sup>RNAi</sup>*—expressed throughout development (n = 20) and *nanos>raptor<sup>RNAi</sup>*—expressed throughout development (n = 20). Means and standard deviations are shown. \*\*\*\* denotes statistical significance (p<0.0001) as determined with Mann-Whitney tests. In all panels, asterisks indicate the hub.

Accumulation of GSC-like cells is not caused by stress or accelerated aging, due to shifting the flies from 18°C to 29°C, as a control RNAi expressed under the control of *nanos^TS* at 29°C did not induce an increase in pMad⁺ germ cells. Similarly, raising and maintaining *nanos^TS*>*mTor^RNAi* flies at 18°C for four weeks, which blocked induction of RNAi expression, also did not result in an increase in pMad⁺ germ cells (**S2V Fig**). Finally, depletion of mTor in 4–16 cell spermatogonial cysts throughout development did not produce any obvious phenotype in the testis (**S2W and S2X Fig**), in agreement with the observation that mTor::GFP expression starts to decrease in cells that typically express *bag of marbles* (*bam*) (**Fig 1B**). Altogether, our results indicate that downregulation of mTORC1 activity in adult male GSCs and early germ cells causes a block or delay in differentiation at the earliest steps, leading to the accumulation of germ cells harboring GSC-like features, including spherical fusomes and high BMP signaling.

## mTor depletion in GSCs and early germ cells leads to the accumulation of dysfunctional autolysosomes

Staining testes with antibodies against the germ cell marker Vasa revealed the presence of large areas devoid of Vasa staining in the cytoplasm of mTor-depleted germ cells in all testes examined (**Fig 3A–3C**, arrowheads). Similar observations were made using additional, independent RNAi lines targeting mTor (**S2D, S2E and S3A Figs**), as well as upon targeting Raptor (**S3B Fig**), indicating that this phenotype correlates with mTORC1 downregulation in germ cells. By contrast, Vasa staining was uniform in the cytoplasm of spermatogonia when mTor was depleted only in 4–16 cell cysts (**S2X Fig**).

Given the multiple roles of mTORC1 in repressing autophagy and lysosomal function (reviewed in [50, 51]), we assessed whether these areas devoid of Vasa expression correspond to autophagic structures. Testes in which mTor was depleted from GSCs and early germ cells were stained with Lysotracker Red, a membrane-permeable vital dye that labels acidic compartments and is used to identify lysosomes [52]. In control testes, Lysotracker stained groups of spermatogonia, and the dye was distributed diffusely throughout the germ cells (**Fig 3C**), a pattern previously characterized in adult testes and corresponding to naturally occurring spermatogonial cell death [53, 54]. By contrast, in mTor-depleted germ cells, Lysotracker consistently labeled cytoplasmic regions devoid of Vasa staining (**Fig 3D**, arrowheads), suggesting that these may be lysosomes. We did not observe diffuse Lysotracker staining as is observed in dying spermatogonia, supporting our hypothesis that the accumulation of early germ cells is the result of a defect in differentiation, rather than cell death in spermatogonia.

To determine whether these Lysotracker positive structures are autolysosomes, in which autophagic cargos are degraded, we assessed the expression of an mCherry-tagged *Atg8a* transgene, placed under the control of the *Atg8a* promoter [55], which permits visualization of all autophagic structures [52]. In control testes, mCherry-Atg8a formed small puncta that were primarily detected in hub cells and occasionally in CCs and spermatogonia, as previously reported [56] (**S3C Fig**). By contrast, mCherry-Atg8a formed large structures that filled the cytoplasmic regions devoid of Vasa staining upon depletion of mTor (**S3D Fig**). Staining positive for both Lysotracker and Atg8a is indicative of autolysosomes. Moreover, as mCherry-Atg8a is expressed under the control of *Atg8a* promoter, our observations indicate that mTor depletion induces an upregulation of Atg8a expression in germ cells, consistent with recent reports showing that mTORC1 negatively regulates the expression of autophagy related genes, including *Atg8a*, by affecting mRNA processing [57, 58].

The autolysosomes that form in germ cells upon Tor depletion appeared strikingly large in comparison to the ones observed in control germ cells (**Fig 3F**), which could reflect a defect in

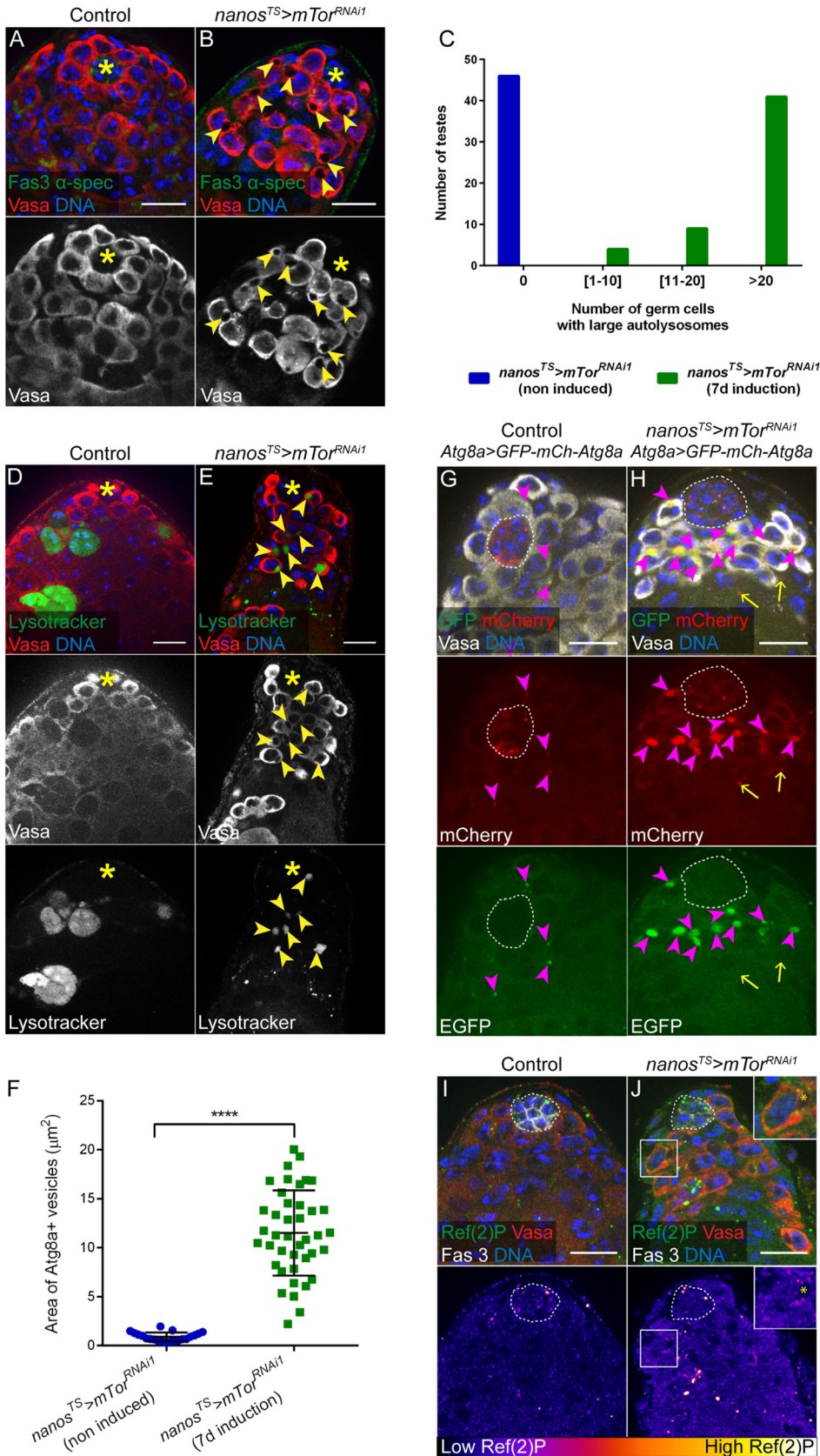

**Fig 3. mTor depletion in GSCs/early germ cells leads to the accumulation of dysfunctional autolysosomes.** (A-B) Control (A) and mTor-depleted (B) testis tips with Vasa, α-spectrin and Fas3 stainings. Arrowheads signal large cytoplasmic areas devoid of Vasa staining in mTor-depleted germ cells. (C) Quantification of the number of testes with 0, 1–10, 11–20 or more than 20 germ cells with large cytoplasmic regions devoid of Vasa staining (later characterized as autolysosomes), in non-induced control flies (n = 46) and flies in which mTor depletion is induced in adult GSCs and early germ cells (n = 54). (D-E) Control (D) and mTor-depleted (E) testis tips with Lysotracker Red and Vasa stainings. Arrowheads signal Vasa-negative and Lysotracker-positive cytoplasmic areas in mTor-depleted germ cells. (F) Area of Atg8a+ vesicles (in $\mu m^2$) in non-induced control flies (n = 21 vesicles) and flies in which mTor depletion is induced in adult GSCs and early germ cells (n = 41 vesicles), carrying the *pAtg8a-mCherry-Atg8a* transgene. **** denotes statistical significance (p<0.0001) as determined with a two-tailed unpaired t-test. (G-H) Control (G) and mTor-depleted (H) testis tips ubiquitously expressing the EGFP-mCherry-Atg8a autophagic flux sensor. Magenta arrowheads signal yellow (EGFP and mCherry-positive) Atg8a puncta in germ cells and the yellow arrows point to red (mCherry-positive and EGFP-negative) Atg8a puncta in cyst cells. (I-J) Control (I) and mTor-depleted (J) testis tips stained with Ref(2)P, Vasa and Fas3. The inset in J is a magnification of the framed germ cell, showing Ref(2)P aggregates in the cytoplasm, outside of the Vasa-devoid cytoplasmic are denoted with an asterisk. Lower panels show Ref(2)P-associated fluorescence intensity represented as a color gradient with the lowest intensity in black and the brightest intensity in white. Testes shown in panels (A-E) and (I-J) are from flies of the genotype *nanos-GAL4/+;UAS-mTor^{TRiP-HMS00904}/tub-GAL80^{ts}* and testes in panels (G-H) are from flies of the genotype *nanos-GAL4/pAtg8a-EGFP-mCherry-Atg8a;UAS-mTor^{TRiP-HMS00904}/tub-GAL80^{ts}*. Control corresponds to non-induced conditions (flies maintained at 18°C) and *nanos^{TS}>mTor^{RNAi}* corresponds to induced conditions (flies raised at 18°C and shifted to 29°C for 7 days at the adult stage). The hub is indicated with an asterisk in (A-D) and is delimited by white dotted lines in (E-H). For all images, scale bar: 15μm.

autophagic flux and lysosomal degradation. To test this hypothesis, we monitored the expression of a dual-tagged *GFP-mCherry-Atg8a* transgene, expressed under the control of *Atg8a* regulatory sequences [59]. GFP-mCherry-Atg8a appears yellow in autophagosomes; upon fusion with lysosomes, GFP fluorescence is more rapidly quenched by the acidic environment in the autolysosome, such that tagged Atg8a appears red. Therefore, we predicted that we would observe red and yellow puncta in control testes, but if autophagic flux is interrupted, the puncta will remain yellow. In control testes, GFP-negative and mCherry-positive Atg8a puncta were primarily observed in hub cells (**Fig 3G**, hub delimited by dashed line), indicating that autophagasomes present in these somatic cells are able to acidify, as previously reported [60]. The rare Atg8a puncta observed in germ cells in control testes appeared yellow (**Fig 3G**, arrowheads), implying that either these puncta correspond to autophagosomes that have not fused with lysosomes or that autolysosomes in wild-type germ cells do not properly acidify. This suggests that autophagy may not be active in germ cells under homeostatic conditions, in agreement with previous findings [56]. In testes from adult flies in which mTor was depleted in GSCs and early germ cells, red Atg8a puncta were observed in the hub (**Fig 3H**, circled) and in some CCs (**Fig 3H**, arrows). Remarkably, large yellow Atg8a vesicles accumulated in the cytoplasm of germ cells (**Fig 3H**, arrowheads). These data suggest that autolysosomes that form upon targeted depletion of mTor in adult early germ cells do not properly acidify. A ubiquitously expressed GFP-LAMP fusion protein [61, 62], which is targeted to lysosomes where hydrolases normally degrade GFP, was also detected in these vesicles (**S3E and S3F Fig**), further supporting the finding that acidification either does not take place or is not efficient.

Ref(2)P, or p62 in mammals, is an intracellular receptor that binds ubiquitinated proteins and targets them for selective autophagic degradation [63]. As such, accumulation of Ref(2)P is commonly used as a readout for defects in autophagy [52]. Upon depletion of mTor in GSCs and early germ cells, we observed an increase in Ref(2)P aggregates in germ cells (**Fig 3I and 3J**). This is in agreement with a defect in autophagic flux leading to the accumulation of undegraded proteins marked with Ref(2)P. Therefore, consistent with a known role for mTORC1 for repressing autophagy, our observations indicate that mTor depletion leads to an increase in autophagic structures in GSCs and early germ cells.

## Down-regulation of mTor in adult GSCs and early germ cells induces ectopic TOR activity in the soma and premature differentiation of cyst cells

When verifying that pS6 was efficiently lost in adult GSCs/early germ cells upon mTor depletion, a concomitant increase in pS6 signal was observed in the adjacent somatic CC compartment (S1G Fig). Indeed, the number of pS6$^+$ early CCs, identified by the expression of Traffic Jam (TJ), was significantly higher when mTor was depleted in GSCs/early germ cells (Fig 4A–4C). This increase in pS6 reflects *bona fide* mTORC1 activity, as it is lost upon rapamycin

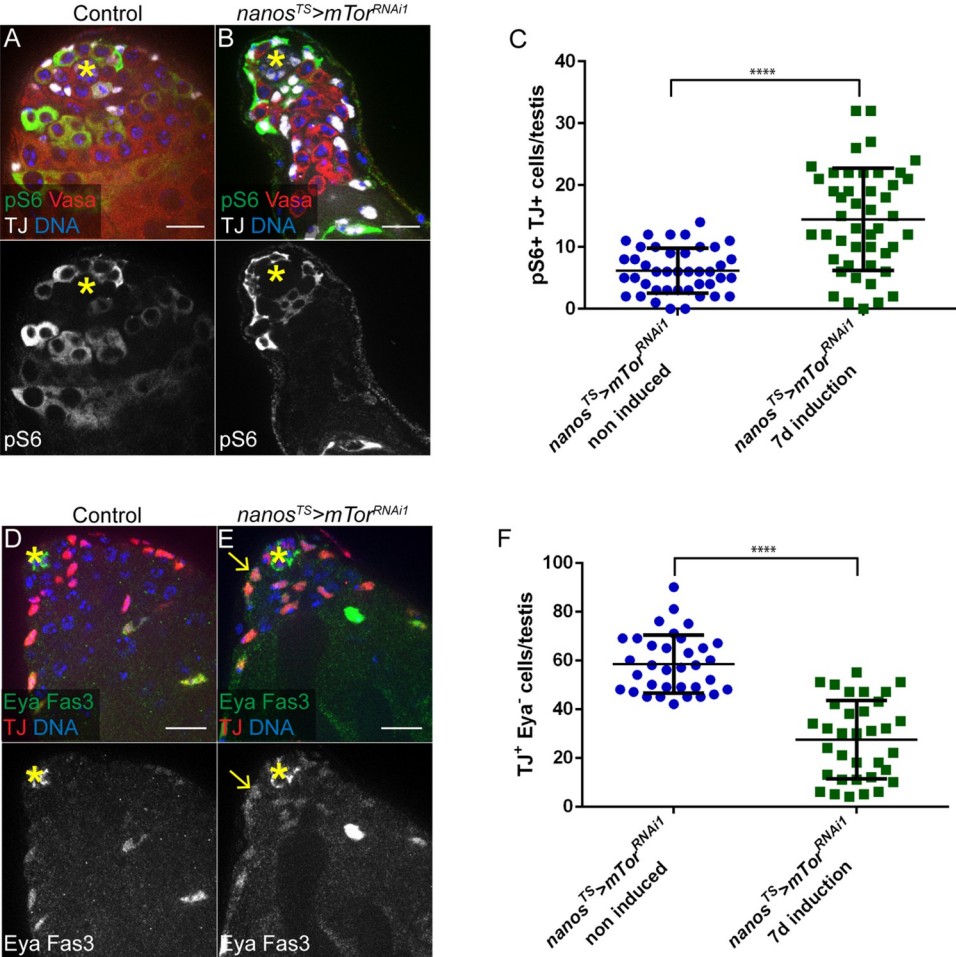

**Fig 4. Down-regulation of mTor in adult GSCs/early germ cells induces ectopic TOR activity in the soma and premature differentiation of cyst cells.** (A-C) Down-regulation of mTor in germ cells induces a concomitant increase in TOR activity in neighboring cyst cells. (A-B) Control (A) and mTor-depleted (B) testis tips with pS6, Vasa and TJ stainings. (C) Number of pS6- and TJ-double positive cells per testis tip in the indicated conditions: *nanos$^{TS}$>mTor$^{RNAi1}$*—non induced (n = 42) and *nanos$^{TS}$>mTor$^{RNAi1}$* - 7days induction (n = 46). Means and standard deviations are shown. **** denotes statistical significance (p<0.0001) as determined with Mann-Whitney tests. (D-F) Down-regulation of mTor in germ cells leads to a loss of early cyst cells. (D-E) Control (D) and mTor-depleted (E) testis tips with Eya, TJ and Fas3 stainings. The arrow in (E) points to an example of Eya-positive cyst cell in proximity to the hub. (F) Number of early cyst cells (TJ positive and Eya negative cells) per testis tip of the indicated conditions: *nanos$^{TS}$>mTor$^{RNAi1}$*—non induced (n = 34) and *nanos$^{TS}$>mTor$^{RNAi1}$* - 7days induction (n = 34). Means and standard deviations are shown. **** denotes statistical significance (p<0.0001) as determined with Mann-Whitney tests. Testes shown in panels (A-B) and (D-E) are from flies of the genotype *nanos-GAL4/+;UAS-mTor$^{TRiP-HMS00904}$/tub-GAL80$^{ts}$*. Control corresponds to non-induced conditions (flies maintained at 18˚C) and *nanos$^{TS}$>mTor$^{RNAi1}$* corresponds to induced conditions (flies raised at 18˚C and shifted to 29˚C for 7 days at the adult stage). The hub is indicated with an asterisk. For all images, scale bar: 15µm.

treatment (**S4A and S4B Fig**). We also observed a similar increase in TOR activity when *raptor* was depleted in early germ cells, throughout development (**S4C and S4D Fig**). Therefore, downregulation of mTORC1 activity in the germline induces non-autonomous activation of mTORC1 in the neighboring CCs.

mTORC1 activity is required for differentiation of CCs, and CC-specific downregulation of mTORC1 leads to an accumulation of early CCs [21, 22]. Thus, we monitored the differentiation of the CC lineage in testes in which mTor is depleted in adult early germ cells. In the somatic lineage, the transcription factor TJ is expressed in CySCs and CCs that surround spermatogonia undergoing TA divisions [64], while late-stage CCs in contact with differentiating spermatocytes express high levels of the transcription factor Eyes absent (Eya) [65]. Therefore, the number of early CCs can be quantified by counting the number of cells expressing TJ with low or no Eya [56]. Upon depletion of mTor in GSCs and early germ cells, a marked decrease in the number of TJ+ Eya- early CCs was observed (**Fig 4F**). In addition, Eya+ cells with large nuclei were observed closer to the testis tip (**Fig 4D and 4E**), including in contact with the hub (**Fig 4E**, arrow). Of note, the Hedgehog target gene Patched (Ptc), which is also used as a marker for CySCs [66], was expressed in somatic cells directly in contact with the hub (**S4E and S4F Fig**, arrowheads), suggesting that Hedgehog signaling is not affected. In addition, proliferating cyst cells were observed in contact with the hub upon germline depletion of mTor, in proportions similar to control testes (**S4G–S4I Fig**). Although Eya expression may not be sufficient to induce CC differentiation [65], the loss of early CCs, together with the observation that more mature CCs are located at the testis tip, suggest that CCs may differentiate prematurely or at a higher rate upon downregulation of TOR in germ cells and the subsequent non-autonomous upregulation of TOR in the soma. Strikingly, mature CCs are found adjacent to mTor-depleted germ cells that exhibit GSC-like features (**Fig 2**), indicating that somatic and germline differentiation appear to be uncoupled. Rapamycin treatment led to a complete loss of Eya expression (**S4J and S4K Fig**), both in control testes, as previously reported [21, 22], and in testes expressing *mTor^RNAi* in GSCs and early germ cells (**S4L and S4M Fig**). Hence, the premature expression of Eya observed in this condition is dependent on mTORC1 activity.

## Discussion

Our findings highlight an essential role for the TOR pathway in regulating GSC behavior in the *Drosophila* testis. Germ cell specific RNAi-mediated depletion of the mTor kinase in testes of adult males induced an accumulation of germ cells expressing markers of GSCs, suggesting a delay or arrest in germ cell differentiation. In addition, cells depleted for mTor exhibited abnormally elevated BMP signaling, upregulated autophagy and an accumulation of dysfunctional autolysosomes. Depletion of mTor in GSCs and early germ cells also led to non-autonomous activation of mTORC1 in neighboring CCs, which correlated with premature differentiation of the CC lineage. Previously, mTor has been shown to play a role in regulating proliferation and differentiation in a variety of stem cell systems, including the fly female germline [17]. Our data indicate that mTor plays a similar role in early germ cells of the fly testis.

Using pS6 as readout, we found that mTORC1 was active in a subset of GSCs and early germ cells that express mTor::GFP. Although it remains unclear whether Tor::GFP expression in an accurate predictor of mTor activity in the germline, our data suggest that mTor kinase may be differentially expressed and, thus, not all early germ cells may be able to respond to upstream signals modulating TOR activity. Of note, p4E-BP, which is commonly used as a reporter for mTor activity, was not readily detected in germ cells, although overexpression of

4E-BP in germ cells did result in p4E-BP staining (**S1J Fig**), suggesting that 4E-BP may not be present at adequate levels in male germ cells under homeostatic conditions to be a primary target. Both pS6 and p4E-BP were detected in early CCs [21], consistent with the well-characterized role for mTor in the somatic lineage in the testis (**Fig 4A, 4C**, and **S1H Fig**). Therefore, our observations highlight the importance of assessing the phosphorylation of different targets as readouts of TOR activity.

Similar to depletion of mTor in GSCs and early germ cells, germline-specific depletion of Raptor and ubiquitous inhibition of mTORC1 through rapamycin feeding led to an accumulation of germ cells harboring GSC-like features, including spherical fusomes and high levels of nuclear pMad (**Fig 2**). Interestingly, mTORC1 was shown to be a negative regulator of BMP signaling in other systems [67–69]. For example, in the female *Drosophila* germline, downregulation of BMP signaling upon hyperactivation of TOR in GSCs contributes to premature differentiation [18]. In mammals, mTORC1 downregulates BMP signaling to promote hair follicle stem cell activation [70] and the transition from early-stage oligodendrocyte precursor cells to immature oligodendrocytes in the central nervous system [69]. Thus, down-regulation of BMP signaling appears to be a conserved mechanism through which mTORC1 regulates the transition into a differentiation program. The mechanisms involved in this process remain to be elucidated.

Depletion of mTor led to an accumulation of dysfunctional autolysosomes, together with Ref(2)P$^+$ aggregates, suggesting that TOR may play an important role in regulating proteostasis in GSCs and early germ cells via repression of the autophagy pathway. Interestingly, BMP signaling has been shown previously to be a positive regulator of autophagy [71, 72]. Therefore, it is possible that, in addition to the direct effects of mTor depletion on autophagy, upregulation of the BMP pathway in mTor-depleted germ cells also induces autophagy, leading to unusually large autolysosomes that appear to be stalled. Our laboratory recently reported that RNAi-mediated depletion of Atg proteins in GSCs and early germ cells does not have a significant impact on GSC maintenance or germline differentiation [56]. Therefore, we hypothesize that autophagy may be actively repressed by TOR signaling in GSCs and early germ cells under homeostatic conditions.

We also found that downregulation of mTORC1 in GSCs and early germ cells led to increased mTORC1 activity in neighboring CCs, which correlated with a loss of early CCs through premature differentiation. This is reminiscent of studies in mice reporting that caloric restriction (CR) leads to inhibition of mTORC1 in the Paneth support cells of intestinal crypts and a concomitant increase in mTORC1 activity in adjacent intestinal stem cells, promoting their proliferation [73, 74]. Of note, in the intestine, CR induces an increase in pS6 but does not affect p4E-BP, further highlighting the importance of assessing multiple readouts for TOR activity. The mechanisms underlying the metabolic crosstalk between soma and germ line in the testis have not been thoroughly characterized. One possibility is that upon mTORC1 inhibition, metabolites released from germ cells, such as degradation products resulting from autophagy, act as signals to activate mTORC1 in neighboring cells. Indeed, autophagy can act as an upstream regulator of TOR [51], and it was previously shown that upon extended inhibition of mTORC1, the recycling of cellular macromolecules through autophagy can stimulate a feedback mechanism leading to mTORC1 reactivation [75]. A similar pathway was also described in the context of paligenosis, a process by which mature cells de-differentiate and acquire stem cell features in response to injury. Following injury, mTORC1 is first inhibited, leading to upregulation of the autophagic and lysosomal machineries, followed by subsequent reactivation of mTORC1 in a lysosome-dependent manner [76, 77]. In our model, mTORC1 cannot be reactivated in GSCs and early germ cells as the mTor kinase is depleted in these cells. Therefore, degradation products generated by autophagy in mTor-depleted germ cells

could be secreted via lysosomal exocytosis or other forms of non-conventional secretion that are dependent on autophagy [78] and signal to activate mTORC1 in neighboring CCs. In line with this hypothesis, a recent study showed that mTORC1 inhibition stimulates unconventional protein secretion [79].

Importantly, the fact that germline differentiation is delayed or arrested, while somatic cell differentiation appears to be accelerated, indicates that there is uncoupling of the differentiation of these two lineages, which is reminiscent of the phenotypes induced by disruption of EGFR signaling. The EGF ligand Spitz is produced by GSCs to activate EGFR signaling in adjacent somatic cells [80, 81], and under homeostatic conditions, EGFR activity is required autonomously for CySC maintenance. Similar to mTor depletion in GSCs and early germ cells, loss of function mutations in either *Egfr* [80] or the downstream effector *raf* [82] cause an accumulation of early germ cells, as well as somatic cells expressing differentiation markers, such as Eya, close to the hub [56, 82]. Here, we find that depleting mTor in GSCs causes an activation of TOR in neighboring CCs, which appears to contribute to premature differentiation. In a previous study, we showed that EGFR and TOR signaling act antagonistically in CCs, as EGFR stimulates autophagy to control early CC behavior and TOR suppresses autophagy to allow CC differentiation [56]. It is not known whether EGFR contributes to TOR inhibition in early CCs. Based on our observations, one interesting hypothesis is that TOR could be required for normal production of Spitz by germ cells; if so, mTor depletion in GSCs would alter EGFR signaling in neighboring CCs. However, it may be that metabolites secreted from the mTor-depleted germ cells act together with altered EGFR signaling to disrupt tissue homeostasis in this model.

In summary, our analysis demonstrates that mTORC1 is required for proper and timely differentiation of male GSCs under homeostatic conditions, pointing to a conserved role for mTORC1 in regulating the transition of progenitor cells to a differentiation program in a number of adult stem cell systems [16]. In particular, studies in the mouse testis reported a similar correlation between mTORC1 activity and differentiation within the spermatogonial progenitor pool [83]. In the mouse testis, rapamycin treatment was also shown to block spermatogonial differentiation, leading to an accumulation of undifferentiated spermatogonia [83–85] that seem to harbor enhanced sensitivity to niche cytokines [83], while rapamycin analogs were associated with reversible male infertility in human [86, 87]. Thus, mTORC1 requirement for spermatogenesis and germ cell differentiation appears to be conserved.

## Supporting information

**S1 Fig.** (A-B) Rapamycin treatment leads to a loss of pS6 staining without change in mTor:: GFP expression. *mTor*::*GFP* flies were placed on food containing either ethanol as a control (A) or rapamycin (4mM in ethanol) (B) for 5 days. Images show testis tips with GFP, pS6, Fas3, α-spectrin and DNA stainings. Dotted lines delimitate the contour of the testis shown in (B). (C-D) RNAi-mediated depletion of mTor in germ cells induces a loss of pS6. Images show testis tips from *nanos-GAL4/+;UAS-mTor^{TRiP-HMS00904}/tub-GAL80^{ts}* animals raised at 18˚C and either maintained at 18˚C (C) (Control) or shifted to 29˚C at the adult stage for 7 days prior to dissection (D) (*nanos^{TS}>mTor^{RNAi1}*), with pS6, Fas3, α-spectrin, Vasa and DNA stainings. Arrowheads on panel (D) signal Vasa⁻ pS6⁺ cyst cells. (E-F) Genetically induced hyperactivation of mTORC1 leads to an increase in pS6-positive cells. Images show examples of testis tips from control (E) (*nanos-GAL4:VP16/UAS-white^{TRiP-HMS00045}*) and Iml1-depleted (F) (*UAS-iml1^{TRiP-HMC04806}/+;nanos-GAL4:VP16/+*) animals, with pS6, Vasa, Fas3, α-spectrin and DNA stainings. (G) Quantification of early germ cells with pS6 staining in the indicated conditions: *mTor*::*GFP*—control (n = 5), *mTor*::*GFP* + rapamycin (n = 5), *nanos^{TS}>mTor^{RNAi1}* non

induced (n = 24), $nanos^{TS}>mTor^{RNAi1}$ 7 days induction (n = 27), $nanos>white^{RNAi}$ (n = 8), $nanos>iml1^{RNAi}$ (n = 18). ** (p<0.01) and **** (p<0.0001) denote statistical significance as determined with Mann-Whitney tests. (H-J) Phosphorylation of 4E-BP does not respond to variations in TOR activity in male germ cells. Images show testis tips from control (H) ($nanos$-$GAL4$:$VP16$/$UAS$-$white^{TRiP-HMS00045}$), Iml1-depleted (I) ($UAS$-$iml1^{TRiP-HMC04806}$/+;$nanos$-$GAL4$:$VP16$/+) and 4E-BP-overexpressing (J) ($UAS$-$thor$/+;$nanos$-$GAL4$:$VP16$/+), with p4E-BP, Fas3, Vasa and DNA stainings. (K-L) mTor::GFP expression strongly decreases upon downregulation of mTor in GSCs and early germ cells. Images show examples of testis tips from control (K) ($nanos$-$GAL4$/$tub$-$GAL80^{ts}$;$UAS$-$white^{TRiP-HMS00045}$/$mTor$::$GFP$) and mTor-depleted (L) ($nanos$-$GAL4$/$tub$-$GAL80^{ts}$;$UAS$-$mTor^{TRiP-HMS00904}$/$mTor$::$GFP$) animals raised at 18˚C and shifted to 29˚C for 7 days at the adult stage to induce expression of the RNAi transgenes, with GFP, Fas3, Vasa and DNA stainings. In the lower (L) panel, the testis is contoured with dotted lines. (M) mTor::GFP expression pattern in the first stages of oogenesis. Image shows an example of germarium with one egg chamber from an ovary of a fly carrying the $mTor$::$GFP$ transgene and expressing RFP under the control of $bam$-$GAL4$, stained with an anti-GFP antibody and DAPI. In all panels, asterisks denote the hub. All scale bars: 15μm. (TIF)

**S2 Fig.** (A-B) Depletion of mTor in germ cells with $mTor^{RNAi2}$ induces an accumulation of early germ cells with spherical fusomes. Images show testis tips from $nanos$-$GAL4$/+;$UAS$-$mTor^{TRiP-HMS01114}$/$tub$-$GAL80^{ts}$ animals raised at 18˚C and either maintained at 18˚C (A) (Control) or shifted to 29˚C at the adult stage for 15 days prior to dissection (B) ($nanos^{TS}>m$-$Tor^{RNAi2}$), with Vasa, α-spectrin (α-spec), Fas3, Traffic-Jam (TJ) and DNA stainings. Lower panels show maximum intensity projections of several images spanning the hub region (Z-projections). (C) Quantification of the number of germ cells (Vasa$^+$ cells) in contact with the hub in control testes ($nanos^{TS}>mTor^{RNAi1}$—no induction, n = 39 and $nanos^{TS}>mTor^{RNAi2}$—no induction, n = 21) and in testes with mTor depletion in adult GSCs and early germ cells ($nanos^{TS}>mTor^{RNAi1}$ - 7d induced, n = 46 and $nanos^{TS}>mTor^{RNAi2}$ - 15d induced, n = 43). **** denotes statistical significance (p<0.0001) as determined with Mann-Whitney tests. (D-F) Germline phenotypes induced by mTor depletion in GSCs and early germ cells throughout development with different RNAi ($mTor^{RNAi3}$, $mTor^{RNAi4}$, $mTor^{RNAi5}$). $mTor^{RNAi3}$ induces an accumulation of GSC-like cells with spherical fusomes and large autolysosomes (arrowheads), concomitant with loss of differentiating spermatogonia (D). $mTor^{RNAi4}$ causes a milder phenotype, with germ cells with spherical fusomes several cell diameters away from the hub and large autolysosomes (arrowheads, E). No obvious phenotype was observed with $mTor^{RNAi5}$ in early germ cells (F). Testes are stained with Vasa, α-spectrin, Fas3, DAPI (DNA) (D-F) and Traffic Jam (TJ, in D and F). (G-H) RNAi-mediated depletion of Raptor in germ cells induces an accumulation of early germ cells with spherical fusomes. Images show examples of testis tips from control (G) ($nanos$-$GAL4$:$VP16$/$UAS$-$white^{TRiP-HMS00045}$) and $nanos$-$GAL4$:$VP16$/ $UAS$-$raptor^{TRiP-HMS00124}$ (H) animals, with Vasa, α-spectrin (α-spec), Fas3, Traffic-Jam (TJ) and DNA stainings. Lower panels show maximum intensity projections of several images spanning the hub region (Z-projections). (I-J) RNAi-mediated depletion of mTor in adult germ cells with $mTor^{RNAi2}$ induces an accumulation of pMad$^+$ cells at the tip of the testis. Images show testis tips from $nanos$-$GAL4$/+;$UAS$-$white^{TRiP-HMS00045}$/$tub$-$GAL80^{ts}$ (I) and $nanos$-$GAL4$/+;$UAS$-$mTor^{TRiP-HMS01114}$/$tub$-$GAL80^{ts}$ (J) animals raised at 18˚C and shifted to 29˚C at the adult stage for 15 days prior to dissection, with Fas3, α-spectrin, pMad and DNA stainings. (K-L) Rapamycin treatment induces an increase in pMad-positive germ cells with a spherical fusome. Wild-type $OregonR$ flies were placed on food containing either ethanol as a control (K) or rapamycin (4mM in ethanol) (L) for 7 days. Images show examples of testis tips

with Fas3, α-spectrin, pMad and DNA stainings. (M-N) RNAi-mediated depletion of Raptor throughout development causes an increase in pMad-positive cells with a spherical fusome. Images show testis tips from *nanos-GAL4:VP16/UAS-white*$^{TRiP-HMS00045}$ (M) and *nanos-GAL4:VP16/UAS-raptor*$^{TRiP-HMS00124}$ (N) animals, with Fas3, α-spectrin, pMad and DNA stainings. (O-P) RNAi-mediated depletion of mTor in adult GSCs and early germ cells causes a decrease in expression of the differentiation marker Bam. Images show examples of testis tips from *nanos-GAL4/+;UAS-mTor*$^{TRiP-HMS00904}$/*tub-GAL80*$^{ts}$ animals raised at 18°C and either maintained at 18°C (O) or shifted to 29°C at the adult stage 7days prior to dissection (P), with Bam, Fas3, Vasa and DNA stainings. (Q-R) mTor-depleted single dividing germ cells are observed several cell diameters away from the hub. Images show examples of testis tips from *nanos-GAL4/+;UAS-mTor*$^{TRiP-HMS00904}$/*tub-GAL80*$^{ts}$ animals raised at 18°C and either maintained at 18°C (Q) or shifted to 29°C at the adult stage 7days prior to dissection (R), with phospho-histone H3 (PH3), Traffic Jam (TJ), Vasa and DNA stainings. Images are maximum intensity projections of several images spanning the hub region (Z-projections), which is indicated by dotted lines. (S-U) Depletion of mTor in adult GSCs and early germ cells causes an accumulation of germ cells with activated STAT signaling. Images show examples of testis tips from *nanos-GAL4/+;UAS-mTor*$^{TRiP-HMS00904}$/*tub-GAL80*$^{ts}$ animals raised at 18°C and either maintained at 18°C (S) or shifted to 29°C at the adult stage 7days prior to dissection (T), with phospho-Stat92E (pSTAT), Fas3, Vasa and DNA stainings. The graph in (U) shows the number of pSTAT$^{+}$ germ cells per testis in non-induced controls (n = 28) and upon 7 days induction of mTor depletion in early germ cells (n = 29). Means and standard deviations are shown. **** denotes statistical significance (p<0.0001) as determined with a Mann-Whitney test. (V) Number of pMad-positive germ cells per testis tip of the indicated conditions: *nanos*$^{TS}$>*mTor*$^{RNAi1}$–7 days induction (n = 71), *nanos*$^{TS}$>*mTor*$^{RNAi1}$–28 days non induced (n = 21) and *nanos*$^{TS}$>*white*$^{RNAi}$—10–15 days induction (n = 19). Means and standard deviations are shown. **** denotes statistical significance (p<0.0001) as determined with Mann-Whitney tests. Data from *nanos*$^{TS}$>*Tor*$^{RNAi1}$—induced are the same as in Fig 2G and are shown here for comparison purpose. (W-X) mTor depletion in later stages of spermatogonial differentiation using *bam-GAL4* does not cause any obvious phenotype. Images show testis tips from *bam-GAL4:VP16/UAS-white*$^{TRiP-HMS00045}$ (W) and *bam-GAL4:VP16/UAS-mTor*$^{TRiP-HMS00904}$ (X) animals, with Fas3, α-spectrin, Vasa and DNA stainings. Lower panels show maximum intensity projections of several images spanning the hub region (Z-projections). Asterisks denote the hub. All scale bars: 15μm.

(TIF)

**S3 Fig.** (A-B) mTor-depleted and Raptor-depleted germ cells harbor large cytoplasmic areas devoid of Vasa staining. Images show testis tips from *nanos-GAL4/+;UAS-mTor*$^{TRiP-HMS01114}$/*tub-GAL80*$^{ts}$ animals raised at 18°C and shifted to 29°C at the adult stage for 15 days prior to dissection (A) and *nanos-GAL4:VP16/UAS-raptor*$^{TRiP-HMS00124}$ (B) animals, with Fas3, α-spectrin, Vasa and DNA stainings. Arrowheads point to Vasa-negative structures in the cytoplasm of germ cells. (C-D) Vasa-negative regions in mTor-depleted germ cells contain Atg8a. Images show testis tips from *pAtg8a-mCherry-Atg8a/nanos-GAL4;UAS-mTor*$^{TRiP-HMS00904}$/*tub-GAL80t*$^{s}$ animals raised at 18°C and either maintained at 18°C (C) (Control) or shifted to 29°C at the adult stage for 7 days prior to dissection (D), with Vasa, Fas3 and DNA stainings. Arrowheads in (C) point to Atg8a puncta in cyst cells and germ cells in a control testis. (E-F) GFP-LAMP accumulates in Vasa-negative cytoplasmic structures of mTor-depleted germ cells. Images show testis tips from *tub-GFP::LAMP1/nanos-GAL4;UAS-mTor*$^{TRiP-HMS00904}$/*tub-GAL80t*$^{s}$ animals raised at 18°C and either maintained at 18°C (E) (Control) or shifted to 29°C at the adult stage for 7 days prior to dissection (F), with Vasa, Fas3 and DNA stainings. The

hub is indicated with an asterisk in (A-B) and is delimitated with dotted lines in (C-F). For all images, scale bar: 15μm.
(TIF)

**S4 Fig.** (A-B) Rapamycin treatment prevents the increase of pS6 in cyst cells adjacent to mTor-depleted germ cells. *nanos-GAL4/+;UAS-mTor$^{TRiP-HMS00904}$/tub-GAL80$^{ts}$* flies raised at 18˚C were shifted to 29˚C at the adult stage on food containing either ethanol as a control (A) or rapamycin (4mM in ethanol) (B) for 7 days. Images show testis tips with pS6, Vasa, Fas3, α-spectrin and DNA stainings. (C-D) Depletion of Raptor in germ cells induces an increase of pS6 in neighboring cyst cells. Images show testis tips from *nanos-GAL4:VP16/UAS-white-$^{TRiP-HMS00045}$* (C) and *nanos-GAL4:VP16/UAS-raptor$^{TRiP-HMS00124}$* (D) animals, with pS6, Vasa, Fas3, α-spectrin and DNA stainings. Cyst cells are identified as Vasa-negative and Fas3-negative cells in the testis tip. (E-F) Depletion of mTor in adult germ cells does not affect Patched expression in cyst cells. Images show testis tips from *nanos-GAL4/+;UAS-mTor$^{TRiP-HMS00904}$/tub-GAL80$^{ts}$* animals raised at 18˚C and either maintained at 18˚C (E) or shifted to 29˚C at the adult stage for 7 days prior to dissection (F), with Patched (Ptc), E-Cadherin (Ecad), TJ and DNA stainings. Arrowheads point to Ptc-positive and TJ-positive cells in contact to the hub, identified at CySCs. (G-I) Depletion of mTor in adult germ cells does not affect the ability of CySCs to proliferate. Images show testis tips from *nanos-GAL4/+;UAS-mTor$^{TRiP-HMS00904}$/tub-GAL80$^{ts}$* animals raised at 18˚C and either maintained at 18˚C (G) or shifted to 29˚C at the adult stage for 7 days prior to dissection (H), with EdU, Vasa, Fas3 and DNA stainings. Arrowheads point to EdU-positive and Vasa-negative cells in contact to the hub, identified at CySCs. The graph in (I) shows the number of EdU positive CySCs per testis in the indicated conditions (n = 10 in each condition), "n.s" indicates that the numbers are not significantly different as determined with a Mann-Whitney test. (J-M) Treatment with Rapamycin inhibits Eya expression both in control testes and testes with germline-specific Tor depletion. *nanos-GAL4/+;UAS-white$^{TRiP-HMS00045}$/tub-GAL80$^{ts}$* (J-K) and *nanos-GAL4/+;UAS-Tor$^{TRiP-HMS00904}$/tub-GAL80$^{ts}$* (L-M) flies raised at 18˚C were shifted to 29˚C at the adult stage on food containing either ethanol as a control (J and L) or rapamycin (4mM in ethanol) (K and M) for 7 days. Images show testis tips with Fas3, Eya, Vasa, TJ and DNA stainings. Arrows in (I) points to a cyst cell expressing Eya in proximity to the hub. Asterisks denote the hub. All scale bars: 15μm.
(TIF)

## Acknowledgments

The authors thank Margaret Fuller (Stanford University), Thomas Neufeld (University of Minnesota), David Walker (University of California, Los Angeles), Helmut Krämer (University of Texas Southwestern), the Bloomington Drosophila Stock Center and the Vienna Drosophila Resource Center for fly stocks; Aurelio Teleman (German Cancer Research Center), Jong-kyeong Chung (Seoul National University), Dorothea Godt (University of Toronto), David Walker (University of California, Los Angeles) and the Developmental Studies Hybridoma Bank for antibodies. We acknowledge members of the Jones laboratory for helpful discussions and Jordan Kryza for analysis of the FlyAtlas data, as well as Utpal Banerjee (University of California, Los Angeles) for providing access to laboratory space and reagents.

## Author Contributions

**Conceptualization:** Marie Clémot, Cecilia D'Alterio, D. Leanne Jones.

**Data curation:** Marie Clémot, Cecilia D'Alterio, Alexa C. Kwang.

**Formal analysis:** Marie Clémot, Cecilia D'Alterio, Alexa C. Kwang.

**Funding acquisition:** D. Leanne Jones.

**Investigation:** Cecilia D'Alterio.

**Methodology:** Marie Clémot.

**Project administration:** D. Leanne Jones.

**Supervision:** D. Leanne Jones.

**Validation:** Marie Clémot.

**Writing – original draft:** Marie Clémot, D. Leanne Jones.

**Writing – review & editing:** Marie Clémot, D. Leanne Jones.

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
