## [Decision Letter · Decision Letter 0]

11 Jul 2023

PONE-D-23-15745mTORC1 is required for differentiation of germline stem cells in the Drosophila melanogaster testisPLOS ONE

Dear Dr. Jones,

Thank you for submitting your manuscript to PLOS ONE. After careful consideration, we feel that it has merit but does not fully meet PLOS ONE’s publication criteria as it currently stands. Therefore, we invite you to submit a revised version of the manuscript that addresses the points raised during the review process.

We look forward to receiving your revised manuscript.

Kind regards,

Maria Grazia Giansanti

Academic Editor

PLOS ONE

“Funds to support this study were provided by the UCLA Eli and Edythe Broad Center of Regenerative Medicine and Stem Cell

Research Training Program (MC), and the NIH: AG040288, AG052732, GM135767 (DLJ).”

“The authors thank Margaret Fuller (Stanford University), Thomas Neufeld (University of Minnesota), David Walker (University of California, Los Angeles), Helmut Krämer (University of Texas Southwestern), the Bloomington Drosophila Stock Center and the Vienna Drosophila Resource Center for fly stocks; Aurelio Teleman (German Cancer Research Center), Jongkyeong Chung (Seoul National University), Dorothea Godt (University of Toronto), David Walker (University of California, Los Angeles) and the Developmental Studies Hybridoma Bank for antibodies. We acknowledge members of the Jones laboratory for helpful discussions, Utpal Banerjee (University of California, Los Angeles) for providing access to laboratory space and reagents, the UCLA Eli and Edythe Broad Center of Regenerative Medicine and Stem Cell Research Training Program for support, and NIH: AG040288, AG052732, GM135767 (DLJ).”

“Funds to support this study were provided by the UCLA Eli and Edythe Broad Center of Regenerative Medicine and Stem Cell

Research Training Program (MC), and the NIH: AG040288, AG052732, GM135767 (DLJ).”

Additional Editor Comments:

Your paper has been reviewed by two referees. I think that their reviews are accurate.

The two reviewers some requests that should be addressed in order to accept the article for publications in Plos one.

Reviewers' comments:

Reviewer's Responses to Questions

**Comments to the Author**

1. Is the manuscript technically sound, and do the data support the conclusions?

Reviewer #1: No

Reviewer #2: Yes

2. Has the statistical analysis been performed appropriately and rigorously? 

Reviewer #1: Yes

Reviewer #2: Yes

3. Have the authors made all data underlying the findings in their manuscript fully available?

Reviewer #1: Yes

Reviewer #2: Yes

4. Is the manuscript presented in an intelligible fashion and written in standard English?

Reviewer #1: Yes

Reviewer #2: Yes

5. Review Comments to the Author

Reviewer #1: In this manuscript, Clemot et al. describe analysis of TORC1 activity and function in male germline stem cells (GSCs) and early transit amplifying cells, primarily using a GFP-tagged mTor reporter and RNAi knockdown of TORC1 components and regulators. The authors describe their observations that variable TORC1 expression within these cells, which appears to be concentrated in the earliest germ cells, and that inhibition of TORC1 function in these cells disrupts germ cell progression and autolysosomal activity. The authors identify crosstalk between germ cells and somatic cyst cells that increases TORC1 activity in cyst cells when mTor is inhibited in the germline, which accelerates cyst cell differentiation.

While these observations are interesting and indicate that TORC1 activity is important within these cells, the major findings that TORC1 activity is restricted to the early germ cells and that downregulation of mTor in early germ cells causes a block / delay in differentiation are not fully supported by the data. The major concerns around these points are listed below”

1. The authors use a single reporter of mTor expression, a GFP-tagged mTor transgene. However, as the authors state, this transgene is unable to fully rescue mTor loss of function, suggesting insufficient function. It is unclear why the transgene does not fully rescue, and hence it is unclear whether such a reporter is fully recapitulating normal expression patterns. Because the high expression of mTOR in early germ cells serves as the foundational premise of this paper, additional validation of mTOR expression patten is necessary. RNA in situ to examine mTOR expression might serve this purpose. Also multiple scRNA seq data of the tesits are now available in the field, thus just reanalyzing them to see if mTOR is indeed concentrated in the early germ cells might suffice.

2. The authors use pS6 immunofluorescence as a readout of TORC1 activity, and have very strong controls to confirm this pS6 signal is dependent on TORC1 function (Rapamycin feeding and Iml1 RNAi). However, they observe pS6 sporadically throughout the early germline, it is not correlated with the strength of mTor-GFP, and is detected in both GSCs and transit amplifying cells (possibly as late as 8-cell spermatogonia). While the quantification of pS6 positive germ cells is useful, lumping together all “early germ cells” in this data obscures interpretation, particularly because it is unclear what stages this data includes (GSCs alone? GSCs through gonialblasts? Through 2-, or 4-cell spermatogonia?). More useful would be to separate this data into each stage of transit amplifying divisions to demonstrate both that pS6 is enriched in the earliest cells and that the reduction of pS6 upon mTor RNAi is specific to the earliest cells.

3. The authors attempt to use phosphorylation of 4E-BP as a downstream readout of TORC1 activity, and are successfully able to do so through over expressing 4E-BP to make itself an available target for phosphorylation. However, the selected image in Supplementary Figure 1M demonstrates p4E-BP is almost exclusively in the most differentiated nanos expressing cells (where 4E-BP is being ectopically expressed), and is absent from the GSCs. This result undermines the authors’ claims that TORC1 is most active in GSCs. Again, full quantification of the % of p4E-BP cells among each cell type would strengthen the authors claims. Inclusion of Bam>thor over expression (and the expected lack of p4E-BP) would also help to strengthen this point. The authors do mention that there may be differential S6 and 4E-BP phosphorylation by TORC1, but they do not discuss the apparent more active 4E-BP phosphorylation in more differentiated cells or provide speculation as to what may cause this effect.

4. One of the major pieces of evidence to suggest differentiation of GSCs or GBs into transit amplifying spermatogonia is disrupted in mTor KD is the accumulation of spherical fusomes. However, this finding is presented as a single representative image that is left unclear if it demonstrates a typical or exceptional example. Quantification of the number of germ cells connected by each fusome structure would more clearly demonstrate the accumulation of cells that have failed to enter interconnected transit amplification in the mTor KD condition.

5. The increase in pMad+ cells upon TORC1 inhibition is clearly demonstrated and intriguing. The authors interpret this as ‘delayed differentiation’: if it is the case, STAT activation may still be limited to GSCs. Examining STAT upon TORC1 inhibition will be an important comparison to validate (or modulate) their claim to more accurately describe the impact of TORC1 inhibition. It may also be beneficial to include an additional output of cell stage to detect delayed differentiation, such as testing for an increased portion of nos+ germ cells or a decrease in Bam+ germ cells.

Reviewer #2: In this manuscript, the authors demonstrate a previously unappreciated role for mTOR in differentiation of germ cells in the Drosophila testis. While mTOR was previously shown to be required in female germ cells and somatic cells of the testis, it was unknown whether it played a role in male germ cells. The work presented here fills this gap and shows that germ cell-specific mTOR loss results in blocked differentiation, as well as other phenotypes including accumulation of autolysosomes and mis-coordination between the soma and germline. Although mostly descriptive, this work does contribute valuable new knowledge and understanding of the pathways involved in germ cell differentiation.

The work presented here is sound and rigorous and results are backed up by the use of multiple RNAi lines for mTor and consistent results with knockdown of Raptor as well as Rapamcycin treatment, giving confidence in their findings. Finally, in addition to the main message of this work, that mTOR is involved in germline differentiation, several interesting points are raised that deserve publication, for instance that mTOR may have different targets or readouts in different tissues or that mTOR activity is involved in coupling differentiation between germline and soma. These findings will be important for the community to build on and provide mechanistic insights into germ cell differentiation in the testes.

Addressing the following points would improve the manuscript greatly and ensure that all the conclusions are adequately supported:

1. The authors claim that mTORC1 activity is regulated at least in part by the expression of mTor itself. This argument is based solely on the expression pattern of an mTor::GFP transgene, which does not fully rescue lethality associated with mTor mutation. This lack of full rescue could be due either to the tag preventing the protein from being fully functional, as discussed by the authors, or by the transgene failing to recapitulate endogenous expression fully. In the latter case, the lack of expression of the GFP construct may not reflect endogenous expression, and it is hard to draw very strong conclusions from these data. Could the authors provide further validation, for instance by in situ hybridization, or by examining read counts in the recent Fly Cell Atlas, to show that indeed mTor expression is low in spermatocytes? Failing that, the authors should tone down their conclusions (for example on l. 155: “TOR activity is regulated at least in part through differential expression of the mTor kinase” could be replaced with “TOR activity may be regulated…”), and could consider moving these data to the supplement, as they do not constitute the major part of their findings but instead distract from them.

2. The loss of differentiated germ cells upon mTOR knockdown could be due to a differentiation arrest or alternatively to death of the differentiated cells. The latter possibility is not addressed in the experiments and cannot be formally excluded. While the lack of lysotracker-positive germ cell cysts would suggest that there is no increase in death of differentiated germ cells, it could be that the time point at which lysotracker was examined is too late to capture death that would occur soon after knockdown. This possibility could be acknowledged in the text.

3. A hallmark of GSCs and CySCs is their ability to proliferate – in the case of GSCs, the presence of proliferating cells that are not synchronized with their neighbors, and simply proliferation in the case of CySCs. It would be helpful to assess proliferation upon germline mTOR knockdown, to determine a) whether the ectopic GSC-like cells do indeed behave like GSCs, and b) whether the ectopic Eya expression in somatic cells does indicate a loss of CySCs.

4. The authors show that in control GSCs, autophagosomes stain yellow with GFP-mCherry-Atg8a, suggesting that they do not acidify. Similarly, mTOR knockdowns lead to yellow autophagosomes, albeit larger. Thus, although the data clearly show an accumulation of large autophagosomes upon mTOR knockdown, whether they are dysfunctional is debatable, as the main difference with controls appears to be the size and number rather than function of the autophagosomes.

5. As a minor point, some of the data shown are discontinuous variables (such as number of cells, for example Fig. 2G, 4C,4F) and are therefore by definition not normal. For these, a non-parametric test should be conducted (Mann-Whitney), to avoid making assumptions on data normality. In practice, this will most likely not change the outcomes of whether the changes observed are statistically different.

6. PLOS authors have the option to publish the peer review history of their article (what does this mean?). If published, this will include your full peer review and any attached files.

Reviewer #1: No

Reviewer #2: No

---

## [Author Response · Author response to Decision Letter 0]

26 Sep 2023

We have adjusted formatting, to adhere to Plos guidelines, as requested by the Editor. Detailed responses to specific reviewer comments are included in the Response to Reviewers document.

---

## [Decision Letter · Decision Letter 1]

31 Oct 2023

PONE-D-23-15745R1mTORC1 is required for differentiation of germline stem cells in the Drosophila melanogaster testisPLOS ONE

Dear Dr. Jones,

Thank you for submitting your manuscript to PLOS ONE. After careful consideration, we feel that it has merit but does not fully meet PLOS ONE’s publication criteria as it currently stands. Therefore, we invite you to submit a revised version of the manuscript that addresses the points raised during the review process.

We look forward to receiving your revised manuscript.

Kind regards,

Maria Grazia Giansanti

Academic Editor

PLOS ONE

Journal Requirements:

Additional Editor Comments:

Your revised manuscript has been carefully evaluated by two reviewers. One of the reviewers asked minor changes that should clarify a fundamental aspect of the paper.

Therefore I ask you to address the request of the reviewer so that the manuscript can be accepted for publication in Plos one.

Reviewers' comments:

Reviewer's Responses to Questions

**Comments to the Author**

1. If the authors have adequately addressed your comments raised in a previous round of review and you feel that this manuscript is now acceptable for publication, you may indicate that here to bypass the “Comments to the Author” section, enter your conflict of interest statement in the “Confidential to Editor” section, and submit your "Accept" recommendation.

Reviewer #1: (No Response)

Reviewer #2: All comments have been addressed

2. Is the manuscript technically sound, and do the data support the conclusions?

Reviewer #1: Partly

Reviewer #2: Yes

3. Has the statistical analysis been performed appropriately and rigorously? 

Reviewer #1: Yes

Reviewer #2: Yes

4. Have the authors made all data underlying the findings in their manuscript fully available?

Reviewer #1: Yes

Reviewer #2: Yes

5. Is the manuscript presented in an intelligible fashion and written in standard English?

Reviewer #1: Yes

Reviewer #2: Yes

6. Review Comments to the Author

Reviewer #1: The authors addressed most of comments to the previous round of review. However, their response regarding the use of mTor-GFP as expression reporter is not satisfactory. Subheading of the first result section says ‘Differential expression of mTor correlates with variations in TOR activity throughout germ cell differentiation’ using mTOR-GFP. In the response to the review, the authors claim that mTor-GFP expression pattern is not intended to serve as the foundational premise of this manuscript. It does not matter what data serve as a premise of study, but it is simply a ground rule of science that a claim must be supported by the results. If mTor-GFP transgene does not rescue mTor mutant, how would one know whether the reporter GFP pattern represents endogenous mTor expression pattern? We feel uncomfortable that the argument about what is the premise of the work interferes with more fundamental rule of evidence-based claim. At least the authors should be more transparent about mTor-GFP not functional in the main text, instead of the method section.

Reviewer #2: Thank you for addressing all the comments raised previously. This manuscript will be helpful to the field as a whole. One minor typo you may want to correct in proofing: in the methods, "Rabbit monoclonal anti-phosho-SMAD1/5" should be "phospho".

7. PLOS authors have the option to publish the peer review history of their article (what does this mean?). If published, this will include your full peer review and any attached files.

Reviewer #1: No

Reviewer #2: No

---

## [Author Response · Author response to Decision Letter 1]

5 Jan 2024

We have uploaded a Response to Reviewers that outlines the recent changes to the text and Figures 1 and Supplemental Figure 1 to address the concern that the mTor::GFP line used in the manuscript does not rescue mTor mutant phenotypes and, therefore, does not reflect the full scope of mTor expression and mTORC1 activity. In addition, we corrected a typo highlighted by Rev. 2.

---

## [Decision Letter · Decision Letter 2]

27 Feb 2024

mTORC1 is required for differentiation of germline stem cells in the Drosophila melanogaster testis

PONE-D-23-15745R2

Dear Dr. Jones,

We’re pleased to inform you that your manuscript has been judged scientifically suitable for publication and will be formally accepted for publication once it meets all outstanding technical requirements.

Kind regards,

Gregg Roman, PhD

Academic Editor

PLOS ONE

Additional Editor Comments (optional):

Reviewers' comments:

Reviewer's Responses to Questions

**Comments to the Author**

1. If the authors have adequately addressed your comments raised in a previous round of review and you feel that this manuscript is now acceptable for publication, you may indicate that here to bypass the “Comments to the Author” section, enter your conflict of interest statement in the “Confidential to Editor” section, and submit your "Accept" recommendation.

Reviewer #1: All comments have been addressed

Reviewer #2: All comments have been addressed

2. Is the manuscript technically sound, and do the data support the conclusions?

Reviewer #1: Partly

Reviewer #2: Yes

3. Has the statistical analysis been performed appropriately and rigorously? 

Reviewer #1: Yes

Reviewer #2: Yes

4. Have the authors made all data underlying the findings in their manuscript fully available?

Reviewer #1: Yes

Reviewer #2: Yes

5. Is the manuscript presented in an intelligible fashion and written in standard English?

Reviewer #1: Yes

Reviewer #2: Yes

6. Review Comments to the Author

Reviewer #1: The authors have made it clear about the limitation of the reagents they are using, and at this point, the readers can decide the caveat of the reporter, but other data are reasonably consistent with the model. At this point, it does not make sense to hold off the publication of the paper. It is unfortunate though that the authors did not provide quantification of the existing data we suggested in the first round of the review.

Reviewer #2: (No Response)

7. PLOS authors have the option to publish the peer review history of their article (what does this mean?). If published, this will include your full peer review and any attached files.

Reviewer #1: No

Reviewer #2: No

---

## [Editor Report · Acceptance letter]

12 Mar 2024

PONE-D-23-15745R2 

PLOS ONE

Dear Dr. Jones, 

I'm pleased to inform you that your manuscript has been deemed suitable for publication in PLOS ONE. Congratulations! Your manuscript is now being handed over to our production team.

Kind regards, 

on behalf of

Dr Gregg Roman 

Academic Editor

PLOS ONE